# Towards gender-inclusive innovation: Assessing local conditions for agricultural targeting

**Diana E. Lopez** [1] *, **Romain Frelat** [2], **Lone B. Badstue** [3]

**1** Department of Social Sciences, Knowledge, Technology & Innovation Group, Wageningen University & Research, Wageningen, Gelderland, The Netherlands, **2** Department of Animal Sciences, Aquaculture & Fisheries Group, Wageningen University & Research, Wageningen, Gelderland, The Netherlands, **3** Socioeconomics Programme, Gender Research Unit, CIMMYT, Texcoco, Mexico

* diana.lopezramirez@wur.nl

**Data Availability Statement:** All relevant data are within the paper, on GitHub (https://rfrelat.github.io/GenderClimate.html), and on Zenodo (DOI: 10.5281/zenodo.4395535.

## Abstract

The importance of gender norms in agricultural innovation processes has been recognized. However, the operational integration of these normative issues into the innovation strategies of agricultural interventions remains challenging. This article advances a replicable, integrative research approach that captures key local conditions to inform the design and targeting of gender-inclusive interventions. We focus on the *gender climate* across multiple contexts to add to the limited indicators available for assessing gender norms at scale. The notion of *gender climate* refers to the socially constituted rules that prescribe men's and women's behaviour in a specific geographic location—with some being more restrictive and others more relaxed. We examine the gender climate of 70 villages across 13 countries where agriculture is an important livelihood. Based on data from the GENNOVATE initiative we use multivariate methods to identify three principal components: 'Gender Climate', 'Opportunity' and 'Connectivity'. Pairwise correlation and variance partitioning analyses investigate the linkages between components. Our findings evidence that favourable economic or infrastructure conditions do not necessarily correlate with favourable gender normative conditions. Drawing from two case-study villages from Nepal, we highlight opportunities for agricultural research for development interventions. Overall, our approach allows to integrate local knowledge about gender norms and other local conditions into the planning and targeting strategies for agricultural innovation.

## Introduction

The potential of planned interventions in agriculture to reduce gender and other social disparities remains largely underexplored—with gains from improved agricultural technologies continuously benefitting mostly men and better-off farmers [1, 2]. This, despite equal opportunities for women and girls been recognized as key for agricultural development as well as for the achievement of the United Nations' Sustainable Development Goals (SDGs), particularly SDG5-gender equality.

**Funding:** D.E.L The lead author acknowledges the financial support of the CGIAR Programs on MAIZE and WHEAT (grant number 2100723600). URL for MAIZE: https://maize.org/; URL for WHEAT:https://wheat.org/ The funders had no role in the study design, data collection and analysis, decision to publish, or preparation of the manuscript. The views expressed in the article are those of the authors and not of any organization.

**Competing interests:** The authors have declared that no competing interests exist.

Agricultural research for development (AR4D) interventions and their targeting strategies tend to focus on technical-technological issues such as crop genetic improvement, resource use efficiency, or improved pest and disease management which respond to well-established research that highlights the role of AR4D in food security and poverty alleviation [3, 4]. However, sociocultural, political, economic, and agroecological contexts affect farmers' interactions with agricultural research and technology development shaping the adoption and the distributional effects of AR4D interventions [5–9].

For decades, research has evidenced that gender roles and stereotypes influence agricultural technology uptake and innovation processes—see for instance [1, 6, 10–14]. Furthermore, recent research examining the linkages between agricultural innovation, agency, and gender norms emphasizes the relevance of social norms as part of the enabling—or disabling—context for planned interventions in agriculture and natural resource management [15–17]. Overall, gender norms—understood as the socially constituted rules that frame what is considered typical and appropriate for women and men to be and do in a given society—influence innovation processes and continue to limit women's and girls' abilities to learn about, try out, adopt or adapt, and benefit from new agricultural technologies and practices [16, 18]. Restrictive gender norms can prevent women and youth from participating in and benefitting from agricultural interventions [5, 18–20]. Likewise, interventions poorly informed about gender norms may inadvertently contribute to maintain or intensify social and gender disparities [21, 22].

The acknowledgment that gender norms influence innovation processes at various levels, including in the planned interventions of agricultural research for development organizations [15], has led international development organizations such as the World Bank, to advise that any effort towards creating local enabling environments for agricultural innovation be accompanied by carefully designed targeting approaches to understand local social conditions [8, 23, 24]. To date; however, most of the research that informs the planning and targeting of agricultural interventions focuses on agroecological conditions, market access and/or population information and their influence on technology adoption or the uptake of agronomic practices —see for instance [25–29]. Few studies consider these issues together with political or social conditions [30], and even less bring gender normative concerns into their analysis [12]. This suggests the existence of underlying (untested) assumptions. For one that agroecological, economic, infrastructure and/or population conditions are sufficient to explain agricultural innovation processes making the analysis of sociocultural/gender conditions unnecessary.

This article questions such assumption. Our objective is twofold. First, to empirically test the relationship between gender normative conditions with economic, infrastructure, and population conditions—all of which have been recognized to influence agricultural innovation processes. And second, to develop a replicable, integrative approach that allows to assess local gender and non-gender conditions to inform the design and targeting of AR4D interventions so that—in line with SDG5—they can become gender-inclusive. We test this approach with village-level data to also inform ongoing and future interventions across the sites sampled in this study (i.e., 70 rural villages across 13 countries). Overall, the article aligns with the growing body of system-oriented literature that seeks to integrate sociocultural, agroecological, and economic dimensions into the design, targeting and scaling of agricultural innovations [31–33]. Next, we provide the study's background and introduce the notion of *gender climate*.

### From assessing women's empowerment to assessing gender norms

Many significant advances have been made to assess women's empowerment in agriculture. The Women's Empowerment in Agriculture Index (WEAI) been the most influential to date —in terms of country policies that have ensued based on their research results [34]. WEAI

estimates women's empowerment in five agricultural domains including production, resources, income, leadership, and time use [35]. The index enables between-country and within-country comparisons as well as household and individual analyses. It also assists organizations and projects with measurement tools to assess women's empowerment in agricultural interventions [36]. WEAI has particularly contributed to evidence the relationship between women's empowerment in agriculture and their nutritional outcomes across countries and across target groups, i.e., households, women, and children [37–41] but challenges remain. For instance, the Women's Empowerment in Livestock Index (WELI) was developed in part to balance the needs for context specificity and cross-cultural comparisons in the livestock sector that WEAI could not provide [42]. Likewise, the project-level WEAI (pro-WEAI) ensued as a way to adapt WEAI's indicators to the specific monitoring and impact assessment needs of agricultural interventions [43].

However, and despite efforts to assess women's empowerment in agriculture as well as novel insights on the interlinkages between gender norms, agency, and innovation in rural livelihoods [12, 44–47] there continues to be relatively few methodologies and little guidance on how to assess gender norms across contexts in ways that can inform the targeting and scaling of agricultural interventions without losing sight of local contexts and realities. This is not an exclusive problem of AR4D as it has also been identified in other development areas [48, 49]. For instance, the health and development sector recently created a platform with important—but still limited—resources and tools to assess social norms [48]. In this sense, our study also contributes to promising research in norms assessment, particularly in the agricultural field.

The notion of *gender climate* advanced in this article is inspired by Petesch et al. [47] concept of 'local normative climate' which refers to the prevailing set of gender norms in a village, and how they interact with other dynamics in that same context to differentially shape women's and men's sense of agency and opportunities in their lives. The concept "stresses the highly contextual and fluid processes by which norms shape gender roles and power relations" [47] (p.111). Gender norms are part of broader, formal and informal, social institutions that regulate people's interaction in a given community. However, and unlike other social institutions, gender norms are maintained by internalized and stereotypical beliefs about different types of women and men as well as by broader social expectations that individuals should act in gender-appropriate ways [47]. Due to this, gender norms play a crucial role in shaping men's and women's identity, freedom, voice, power, and access to resources [50]. In this article, *gender climate* refers to the socially constituted (formal and informal) rules that regulate people's daily behaviour in a specific geographic location with some gender norms being more restrictive and others more relaxed. (We discuss the criteria for identifying a climate as 'restrictive' or 'relaxed' in Materials and Methods.)

Petesch et al. use the concept of 'local normative climate' in different works [45–47], most notably to develop a community typology that advances understandings of gender-inclusive agricultural innovation processes and their contribution to empowerment and poverty reduction at village level. They employ comparative case study methods with qualitative data from 79 villages across 17 countries. The typology is designed to inform local and regional AR4D interventions and, to our knowledge, constitutes the only attempt in agri-food systems that compares gender norms at scale. However, the study was a formative multi-site research to explore contextual and other patterning in agency, norms and innovation interactions and did not discuss policy implications such as targeting opportunities.

Our article attempts to capture (at least partially) some of the highly relational, fluid, and contextual properties associated with gender norms. Specifically, we distinguish certain normative issues related to women's physical mobility, decision-making, education, gender-based

violence, and leadership across and within specific geographic locations where agriculture is an important livelihood, and which constitute the basis for *gender climate*.

This article complements WEAI, WELI and Petesch et al. [46] in their efforts to advance a scientific approach that is both contextually sensitive and comparative in scope and in their use of comparative measures drawn from diverse perceptions of local women and men. However, our article differs from these studies in important ways; markedly, in its purpose and research design. We aim to empirically test the relationship between different local conditions considered important for agricultural innovation while providing a replicable approach that integrates knowledge about gender norms into the targeting strategies of AR4D interventions.

## Materials and methods

### (a) Data

We use data from 70 rural villages across 13 countries (Fig 1), collected as part of GENNO-VATE—a comparative, qualitative research initiative focusing on the interlinkages between gender norms, agency, and innovation in agriculture [51]. GENNOVATE features a qualitative comparative case study method that is sensitive to local contexts, standardized semi-structured instruments, and maximum diversity sampling, which together provide a basis for broad patterns to be detected without losing their grounding in local contexts and realities. The analytic approach is informed by a conceptual framework based in feminist and innovation theories and the notion of agency-opportunity structure interactions [16]. In each research village, interdisciplinary local field teams used a standardized package of data collection instruments, including sex-specific focus group discussions with women and men of different socio-economic and age groups, semi-structured individual interviews, and community profile key

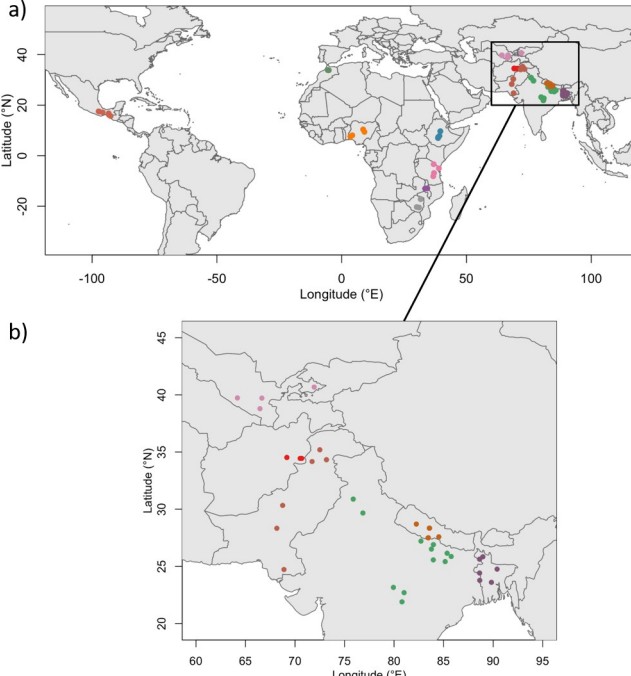

**Fig 1. Map of the study areas with emphasis on Asia.** The colour represents the different villages within country cohorts, in total 70 distributed across 13 countries (Mexico, Morocco, Ethiopia, Tanzania, Malawi, Nigeria, Zimbabwe, Afghanistan, Pakistan, Uzbekistan, Bangladesh, Nepal, and India). Village distribution by region is provided in Table 3.

informant interviews. For more information on the conceptual framework see [16] and [52]. The full methodology and data collection instruments are open access and are available at GENNOVATE website (www.gennovate.org).

We use a sub-set of GENNOVATE data from community profiles and focus group discussions (FGDs) (Table 1). For analysis purposes, the study organizes the village data in two datasets: one focused on gender norms and another one on non-gendered village characteristics including infrastructure development, demographics, and the local economy. An overview of the variables and dimensions derived from these datasets are presented below and in Table 2. The datasets provided the numerical (17) and categorical variables (15) used in the multivariate analysis (MVA).

## (b) Sampling

To the extent possible GENNOVATE selection of villages followed a purposive, maximum diversity sampling [51]. (Specifically, in the case of Afghanistan the sampling was influenced by safety concerns.) The approach sought to capture and describe "central themes that cut across a great deal of variation" [53] (p. 235). The logic was that if a pattern can be uncovered in a large number of varied places there is more confidence (i) in the finding, (ii) that unobserved variables are less important, and (iii) that similar findings likely exist beyond the research sample. Drawing on this approach, GENNOVATE's selection criteria primarily focused on villages

**Table 1. Sources of information.**

| Data Source | Number per village | Name of Instrument | Purpose | Participant selection per village |
|---|---|---|---|---|
| **Key Informant Semi-Structured Interview (KII)** | 2 | Community Profile | To provide social, economic, agricultural, and political background information about the community | *Knowledgeable people recognized by the community (e. g., respected farmers, leaders, government officials, employers, teachers, healthcare workers, etc.)*<br>• 1–2 women<br>• 1–2 men |
| **Focus Group Discussion (FGD)** | 2 | Ladder of Life | • Gender norms, household and agricultural roles<br>• Labour market trends and gender dimensions<br>• Enabling and constraining factors for innovation, and their gender dimensions<br>• The culture of inequity in the village, factors shaping socioeconomic mobility and poverty trends, and their gender dimensions<br>• Intimate partner violence | *Poor adults, ages 30 to 55*<br>• 1 FGD of 8–10 adult women<br>• 1 FGD of 8–10 adult men |
| **Focus Group Discussion** | 2 | Innovation capacities | • Agency<br>• Community trends<br>• Enabling and constraining factors for innovation, and their gender dimensions<br>• Gender norms surrounding household bargaining over livelihoods and assets<br>• The local climate for agriculture and entrepreneurship, and their gender dimensions<br>• Social cohesion and social capital | *Better-off adults, ages 25 to 55*<br>• 1 FGD of 8–10 adult women<br>• 1 FGD of 8–10 adult men |
| **Focus Group Discussion** | 2 | Aspirations of Youth | • Gender norms, practices, and aspirations surrounding education<br>• Enabling and constraining factors for innovation, and their gender dimensions<br>• Women's physical mobility and gender norms shaping access to economic opportunities and household bargaining<br>• Family formation norms and practices | *Older adolescents and young adults, ages 16 to 24*<br>• 1 FGD of 8–10 women<br>• 1 FGD of 8–10 adult men |

Own formulation based on [51, 52].

**Table 2. Overview of variables.**

| # | DIMENSION | CODE | VARIABLE[1] | UNIT | SOURCE OF INFORMATION[2] |
|---|---|---|---|---|---|
| | | | **1st Dataset: Gender norms** | | |
| 1 | Mobility | Perm_mig | Gender difference in permanent migration | % | KII-Community Profile |
| 2 | Mobility | W_Mkt | Share of women selling in local market | % | KII-Community Profile |
| 3 | Mobility | W_All_Jobs | Share of women working | % | KII-Community Profile |
| 4 | Mobility | W_Agri_Jobs | Share of women who take jobs as agricultural workers | % | KII-Community Profile |
| 5 | Mobility | Physical_Mob | Women's physical mobility (women's perception) | Average rating (1 = low, 10 = high) | FGW-Aspirations of Youth |
| 6 | Gender-based violence | Violence | Violence against women (women's perception) | Average rating (1–2 = lower, 3–4 = higher) | FGW-Ladder of life |
| 7 | Decision-making | Inheritance | Women's control over inheritance money (women's perception) | Average rating (1–2 = higher, 3–4 = lower) | FGW-Innovation capacities FGYW-Aspirations of Youth |
| 8 | Decision-making | PF | Change in women's power and freedom to make life decisions such as if or where to work, start or end a relationship, or pursue an education. | Average rating (-5 = decreased, 5 = increased) | FGW- Innovation capacities |
| 9 | Decision-making | Phone_owner | Gender gap in cell phone ownership | % | KII-Community Profile |
| 10 | Decision-making | Ctrl_Wsales[c] | Control of women's agricultural income | 1 = men, 2 = women, 3 = both | KII-Community Profile |
| 11 | Decision-making | Ctrl_Comm[c] | Control over commercial crops or livestock | 1 = men, 2 = women, 3 = both | KII-Community Profile |
| 12 | Decision-making | Ctrl_Subs[c] | Control over subsistence crops or livestock | 1 = men, 2 = women, 3 = both | KII-Community Profile |
| 13 | Governance and leadership | Active_Disc | Share of women active discussants in public meetings/trainings | % | KII-Community Profile |
| 14 | Governance and leadership | Elections[c] | Gender of elected village leader in the last 10 years* | 0 = none, 1 = men, 2 = both | KII-Community Profile |
| 15 | Education | GC_Sec_Edu | Gender gap in secondary school | % | KII-Community Profile |
| | | | **2nd Dataset: Non-gendered village characteristics** | | |
| 16 | Demographics | Pop | Current population | amount | KII-Community Profile |
| 17 | Demographics | Pop_Growth | Population growth (last 10 years) | % | KII-Community Profile |
| 18 | Economy | Farmer_Org | Number of organizations available for local producers in village | amount | KII-Community Profile FGM- Ladder of Life FGW- Ladder of Life FGM- Innovation capacities FGW- Innovation capacities |
| 19 | Economy | Training[c] | Agricultural/non-agricultural job trainings or vocational programs | 1 = yes, 0 = no | KII-Community Profile |
| 20 | Economy | Mkt[c] | Local market/agricultural trade | 0 = no, 1 = daily, 2 = weekly | KII-Community Profile |
| 21 | Economy | Town | Distance to nearest town with government offices | Km | KII-Community Profile |
| 22 | Economy | HHMkt | Share of households that sell own produce in local market | % | KII-Community Profile |
| 23 | Infrastructure | Land | Average land size | Ha | KII-Community Profile |
| 24 | Infrastructure | U_Land[c] | Presence of communal property: unallocated arable land | 1 = yes, 0 = no | KII-Community Profile |
| 25 | Infrastructure | Preschool[c] | Preschool in village | 1 = yes, 0 = no | KII-Community Profile |
| 26 | Infrastructure | Secondary[c] | Lower secondary school in village | 1 = yes, 0 = no | KII-Community Profile |
| 27 | Infrastructure | U_Secondary[c] | Upper secondary school in village | 1 = yes, 0 = no | KII-Community Profile |
| 28 | Infrastructure | Clinic[c] | Health clinic in village | 1 = yes, 0 = no | KII-Community Profile |
| 29 | Infrastructure | Bus[c] | Bus line within half hour walk | 1 = yes, 0 = no | KII-Community Profile |
| 30 | Infrastructure | Electricity[c] | Electricity | 1 = yes, 0 = no | KII-Community Profile |
| 31 | Infrastructure | Internet[c] | Public place with internet access in village | 1 = yes, 0 = no | KII-Community Profile |

*(Continued)*

**Table 2.** (Continued)

| # | DIMENSION | CODE | VARIABLE[1] | UNIT | SOURCE OF INFORMATION[2] |
|---|-----------|------|-------------|------|--------------------------|
| 32 | Infrastructure | Irrigation[c] | Irrigation or water supply for agriculture | 1 = yes, 0 = no | KII-Community Profile |

[1]These variables were used to inform the 'Gender Climate' component (variables 1–15) as well as the two complementary components, 'Opportunity' and 'Connectivity' (variables 16–32).

[2]KII = Key informant interviews; FGM = Focus group discussions with adult men; FGW = Focus group discussions with adult women; FGYW = Focus group discussions with young women.

[c] = Categorical variable.

*Some villages from Nepal and Malawi have not had any elected leader in the last decade due to political instability in these countries.

across and within regions with significant heterogeneity in economic, political, agroecological and sociocultural contexts [51]. A secondary criteria considered the presence of AR4D technical-technological solutions in the villages, such as improved seed varieties or agronomic practices, to ensure villagers' experiences with and exposure to external agricultural interventions [51]. For a full account on the sampling and methodology see [51, 52].

In this article, we use the sampled villages to identify patterns and differences across villages within specific locations (herewith referred to as 'country cohorts'); particularly, about key gender normative conditions that could inform ongoing and future targeting strategies to make them gender-inclusive.

## (c) Dimensions and variables

**1st Dataset: Gender norms.** Considering the historical pre-eminence of men farmers in most agricultural societies [54], and in consequence, their greater access and control over agricultural assets and resources [55], the gender norms dataset of this study is biased towards women: 9 out 15 variables focus only on women rather than on men *vis-à-vis* women. Four of the gender norms' variables (5–8 in Table 2) use ratings to build a comparative story of how individual women perceive certain gender normative issues while also allowing to compare responses between different respondents and groups; overall, providing a relevant overview about these issues in each sampled village—for a similar study see [56]. The 15 variables of the gender norms' dataset are provided in Table 2 (variables 1–15).

Informed by literature on gender norms and women's empowerment [41, 46–49, 57–60], we organized the variables according to five dimensions: mobility, gender-based violence, decision-making, governance and leadership, and education. The criteria for identifying a *gender climate* as 'restrictive' or 'relaxed' was likewise informed by these works; specifically, those of Petesch et al. [46, 47] which identified freedom of mobility and ability to make decisions as key for agricultural innovation. In this article, a *restrictive climate* refers to formal and informal norms that restrict women's mobility, education, decision-making and access to assets and resources. Likewise, it encompasses reported violence against women and lack of leadership positions for women. Conversely, a *relaxed climate* refers to norms that support women's education, leadership, decision-making, access to assets and resources, and women's ability to move freely in the public space. It also suggests no reported violence against women.

The variables that informed the degree of restriction or relaxation across the sampled villages follow. (See the accompanying tutorial for a detailed list on the questions used to derive the 32 variables.)

1. *Mobility*

   This dimension includes five variables. Four out of five variables exclusively look at women's physical mobility (2–5 in Table 2). This dimension captures the differences between men and women in terms of permanent migration (1), share of women selling in the local market (2), share of women working in all types of jobs (3) or only in agriculture (4), and women's own perceptions about their freedom of movement (5). Men and women key informants provided the information for variables 1–4. Variable 5 was calculated using an average rating based on the responses of the young women who participated in the focus group 'Aspirations of Youth'. We asked women to privately rate, out of every 10 women in the village, how many of them were able to move freely on their own in the public spaces. The rating was from 1 to 10: 1 = practically no women move freely on their own in the village; 10 = practically all women move freely on their own in the village.

2. *Gender-based violence*

   This dimension is informed by a single scaled variable which attempts to provide an overview on gender-based violence against women at village level (6 in Table 2). Since women are the main affected party, the variable only considered women's perceptions. We asked adult women who participated in the focus group 'Ladder of Life' to vote privately on the question: to what extent have local women been hit or beaten in their households over the past year? They voted from 1 to 4: 1 = almost never happens here (0 women in 10); 2 = occasionally happens here (1 to 2 women in 10); 3 = regularly happens here (3 women in 10); or 4 = frequently happens here (4 or more women in 10). The variable 'violence' was calculated as an average rating of women's responses to this question.

3. *Decision-making*

   This dimension is made of six variables (8–12 in Table 2). Three variables look at aspects related to both men and women (9,11, 12) and three only consider issues relevant to women (7,8,10). Men and women key informants provided the information for four variables (9–12) including: (sole or joint) decisions over food and cash-crop farming/livestock, control of women's agricultural income, women's control over inheritance money as well as differences between men and women on cell phone ownership. The variable 'inheritance' was informed by two sources: the focus group discussions with adult women ('Innovation Capacities') and the focus group with young women ('Aspirations of Youth'). We asked the two groups of women to privately rate a scenario wherein an innovative/entrepreneurial woman who had recently inherited money, would be able to use this money to purchase a plot of land right near the homestead to expand her vegetable garden. The two-part question was: how easy or difficult will it be for this innovative/entrepreneurial woman to use her inheritance money if her husband would like to use this money to buy himself a motorbike? How easy or difficult will it be for this woman to go ahead and purchase the plot of land without the support from her husband? The women had five options to choose from: 1 = very easy; 2 = easy; 3 = neither easy nor difficult; 4 = difficult; 5 = very difficult. An average rating considering the responses from the two focus groups was calculated. The variable 'PF' which stands for power and freedom, captures women's perception of change in their ability to make life decisions. It was informed by the responses of focus group discussions with adult women ('Innovation capacities'). Women were asked to imagine a five-step ladder wherein each woman was asked to rate their current ability to make important decisions in their lives—including about their working life, starting/maintaining a business or an income-generating activity, their use and control of productive resources, and whether to start or end a relationship. Each step represented a different level of power: step 1 = almost no power or freedom to make decisions; step 2 = only a small amount of power and

freedom; step 3 = power and freedom to make some major life decisions; step 4 = power and freedom to make many major life decisions; step 5 = power and freedom to make most all major life decisions. Women were then asked to locate the position they occupied on the ladder ten years ago, and to reflect upon the reasons for the change. PF was calculated as difference in women's perception between 'now' (at the time of the study) and a decade earlier.

4. *Governance and leadership*
   This dimension is made of two variables (13,14 in Table 2). It focuses on leadership in the village, measured by whether men or women have been elected as village leaders in the last decade (variable 14) as well as women's comfort in speaking in public (variable 13). The latter is represented by the share of women who are actively engaged in public meetings and in trainings. Villages with long-standing internal conflict and no elected leaders are also captured here. The variables were derived from the community profile, which involved both men and women informants.

5. *Education*
   This dimension is exclusively made of a single variable (15 in Table 2). The variable identifies the gender gap in secondary school and was calculated by the difference between men and women to access secondary education. Men and women key informants provided this information.

   **2nd Dataset: Non-gendered village characteristics.** Our study also includes other (non-gendered) village characteristics which are represented by 17 variables (Table 2, variables 16–32). We organized these variables according to three dimensions: demographics, economic conditions, and infrastructure development. This arrangement was informed by the extensive literature on agricultural innovation and technology adoption that highlights these aspects as key for agricultural innovation—see for instance, [3, 11, 30, 31, 61, 62]. World Bank studies, for instance, evidence the importance of investments in infrastructure and markets for innovation capacity [8, 24]. Whereas scholarly works further highlight the relevance of technical knowledge (in the form of agricultural trainings) and of farmer organizations as crucial for innovation [63, 64].

1. *Demographics*
   This dimension includes two variables (16,17 in Table 2): the population estimates during the data collection period as well as the population growth in the last decade. Men and women key informants provided this information.

2. *Economy*
   Five variables are comprised under this dimension (18–22). Four variables were derived from information provided by the men and women who took part in the community profiles: the existence and frequency of a local market (20), the distance to the nearest town with government offices (21), the share of households selling agricultural products in the local market (22), and the presence of (agricultural and/or non-agricultural) job training or vocational programs in the village (19). Variable 18, 'Farmer_Org', represents the total number of organizations supporting local farmers in the village. It was calculated using information from five different sources: community profiles, two focus group discussions with adult men and two with adult women ('Ladder of Life' and 'Innovation Capacities').

3. *Infrastructure*
   This dimensions is made of ten variables (23–32) all of which were informed by key informant men and women involved in the community profiles. Presence of schools, health

clinic, bus line within relative short distance, electricity and internet are accounted within this dimension. Other items such as access to irrigation, presence of unallocated arable land, and average land size are also included here.

### (d) Data analysis

The analysis followed a three-stage process (See Fig 2).

We first built the two datasets and curated them using Microsoft Excel. Once data was deemed coherent and ready to be compared at village level, we explored the datasets using

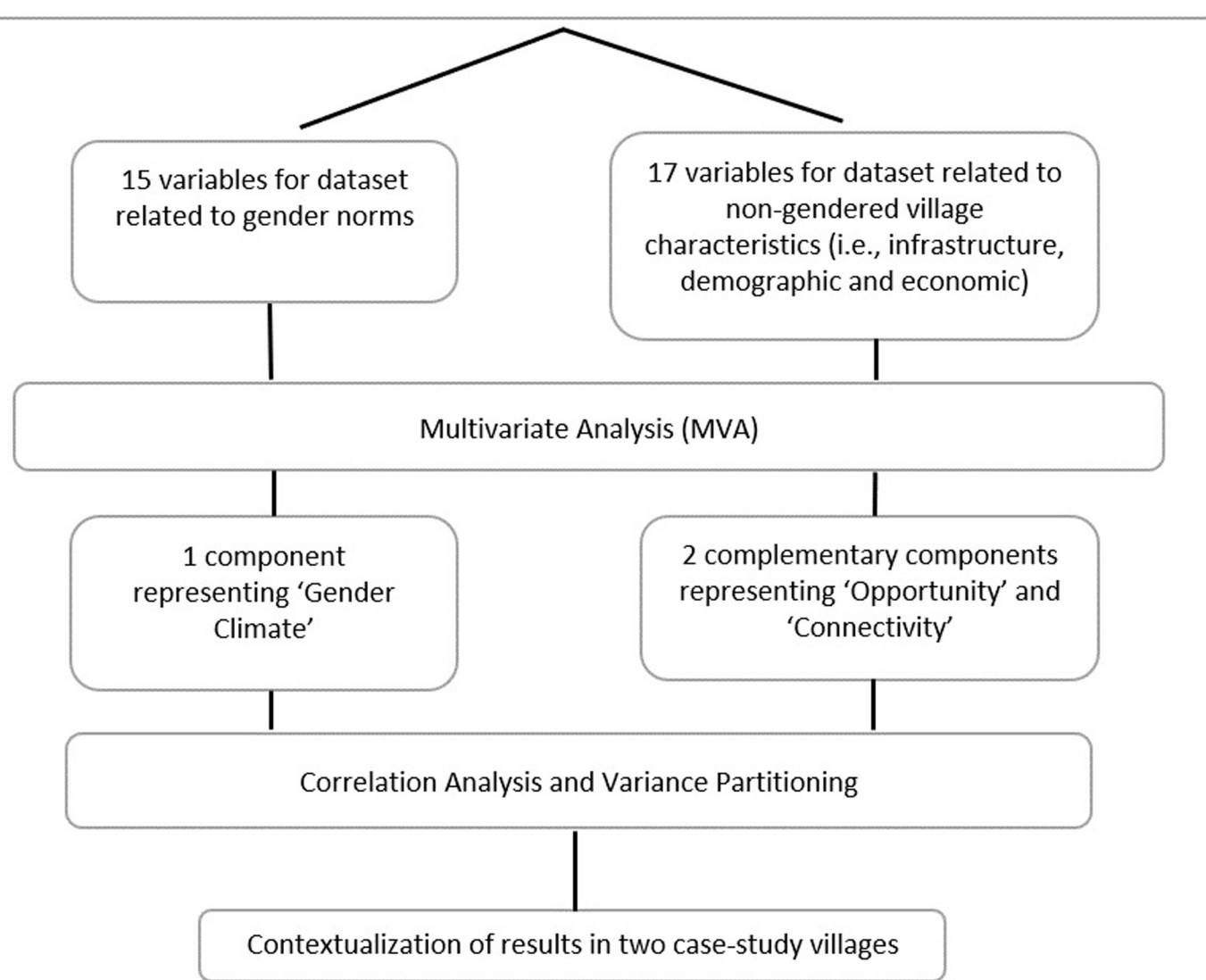

**Fig 2. Schematic representation of the analysis.**

visualization techniques (i.e., histograms, barplots, and pairwise plots). After data exploration we selected variables based on the availability (and consistency) of information. (See tutorial accompanying this article for explanation on variable selection and exploratory analysis.) We then conducted two separate MVA. We used the Hill and Smith method [65] which—although similar to Principal Component Analysis (PCA)—further allows analysis with both categorical and numerical variables [66]. This method helped to simplify the large number of variables into Principal Components (PCs) while maintaining the maximum of the variability in the datasets. MVA provided PCs with 'loadings' for the variables and 'scores' for the villages. We selected the number of PCs based on the distribution of eigen values—representing the percentage of additional variance explained by the successive PCs. We retained PCs that explained the highest amount of variance. Based on the loadings, we interpreted the PCs and named them according to the loadings of the variables (i.e., 'Gender Climate', 'Opportunity', and 'Connectivity').

During the second analysis stage, pairwise correlation analysis and variance partitioning [67] were used to investigate the linkages between the gender climate and the two complementary components across villages. We used variance partitioning to understand the relative influence (i.e., explanatory power) of the two complementary components and the country context *vis-à-vis* the gender climate. All statistical analyses and derived figures were conducted in the programming environment R 3.6 [68]. MVA were carried out with the package ade4 [69]; whereas variance partitioning was calculated with the vegan package [70]. The dataset used in this study as well as a tutorial explaining the MVA and variance partitioning are available online at https://rfrelat.github.io/GenderClimate.html (DOI 10.5281/zenodo.4395534).

Finally, we used qualitative information from the focus group discussions and community profiles to conduct a deeper analysis on the country cohort that reported the most relaxed climate of the dataset, i.e. the Nepal cohort. Particularly, we looked at two case-study villages from this cohort with similar 'Opportunity' and 'Connectivity' but which exhibited large variability in terms of 'Gender Climate'.

## Results

The MVAs provided a PC for (1) 'Gender Climate', which explained 18% of the variability of the gender norms dataset; and (2) two complementary principal components explaining 27% of the non-gendered village characteristics dataset, i.e., 'Opportunity' (14%) and 'Connectivity' (13%).

### (a) Gender climate

The PC of gender climate (henceforth GC) showed high values for country cohorts with a more relaxed gender climate, and low values for country cohorts with a more restrictive gender climate (Fig 3). The degree of restriction or relaxation varied between and within country-cohorts. However, the analysis indicated significant variables that allowed to identify a GC as more or less restrictive or relaxed. The most important categorical variables contributing to a relaxed GC were women's control over their own agricultural income as well as women's (sole) decision-making over commercial crop and livestock (corresponding to only one village in Nepal, NP1). The most significant among numerical variables were women's power and freedom to make life decisions and women's freedom of mobility. The presence of women sellers in local markets, active female discussants in public meetings and trainings, cell phone ownership, as well as women working for pay (both in agricultural and non-agricultural jobs) were also important. Overall, a relatively more relaxed GC across country cohorts appeared to be associated with three normative dimensions: decision-making, mobility, and governance &

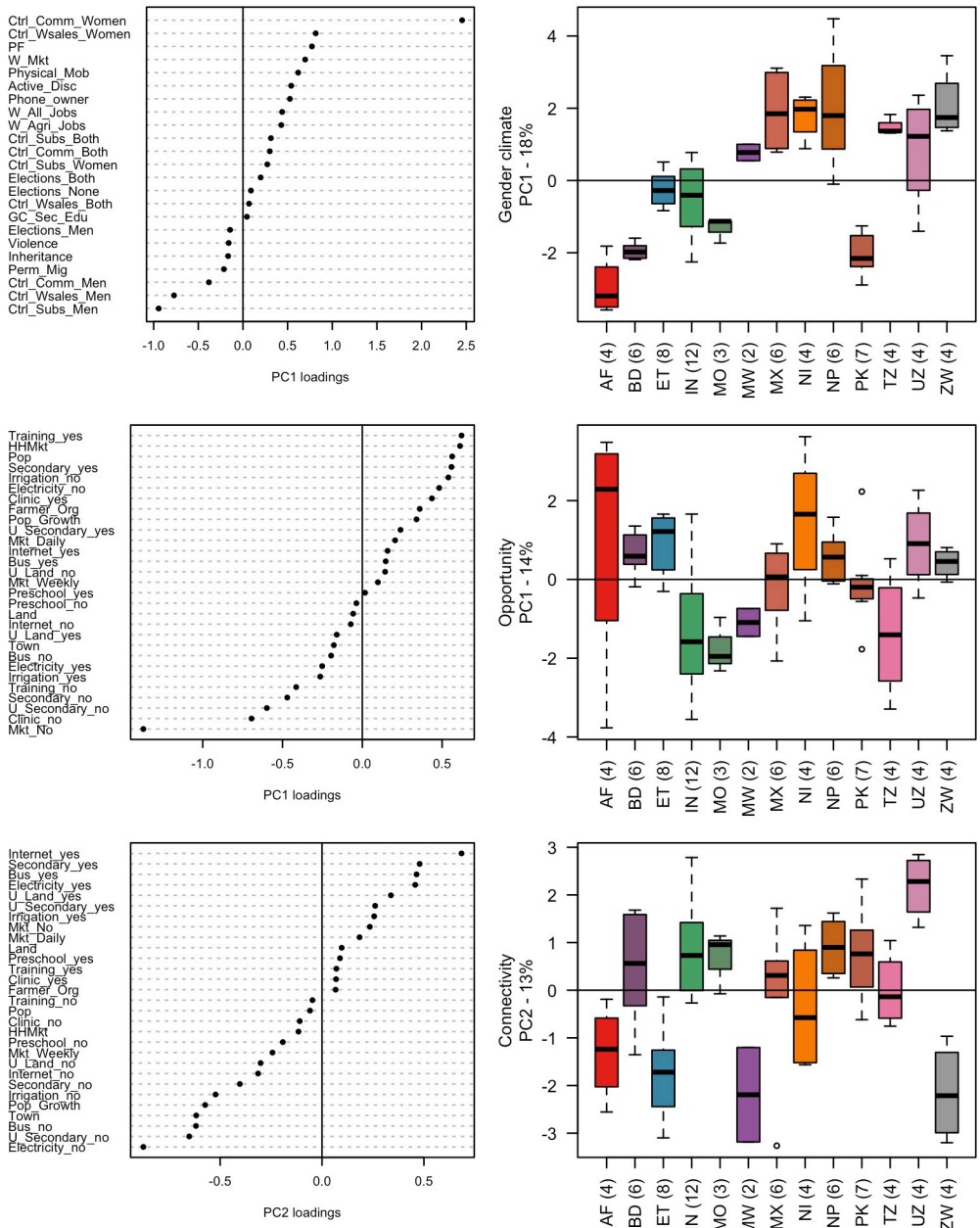

**Fig 3. Loadings of variables and village scores according to 'Gender climate', 'Opportunity', and 'Connectivity'.**
(A)The left column represents the loadings of the variables. (B) The right column represents the distribution of the village scores per country cohort (two letter code), the number of sampled villages per country cohort is shown in parentheses. The colour represents the different villages within country cohorts. (C) The first row is the 'Gender climate' component derived from 15 variables. (D) The second and third rows are the complementary components ('Opportunity' and 'Connectivity') derived from 17 variables.

leadership—in that order. Variables associated with a more restrictive GC included decisions over subsistence crops and livestock made only by men, men's control over women's agricultural income, and decisions over commercial crops and livestock made solely by men.

On average, the more restrictive GC were found in study villages from Afghanistan, Pakistan, and Bangladesh whereas villages in Nepal, Mexico, Nigeria, and Zimbabwe exhibited

**Table 3. Scores of villages according to Gender climate, Opportunity, and Connectivity.**

| Country | Region[*] | Village Code[**] | Gender Climate | Opportunity | Connectivity |
|---|---|---|---|---|---|
| Afghanistan (AF) n = 4 | Kabul | AF1 | -3.42 | 2.90 | -0.19 |
| | Nangarhar | AF2 | -1.82 | 3.48 | -2.55 |
| | Kabul | AF3 | -2.97 | -3.77 | -0.98 |
| | Nangarhar | AF4 | -3.58 | 1.68 | -1.50 |
| Bangladesh (BD) n = 6 | Mymensingh | BD1 | -2.08 | 0.38 | 1.11 |
| | Dhaka | BD2 | -1.88 | -0.19 | -0.33 |
| | Rangpur | BD3 | -2.19 | 1.13 | 1.59 |
| | Rangpur | BD4 | -1.60 | 1.36 | 1.68 |
| | Khulna | BD5 | -2.15 | 0.75 | 0.02 |
| | Rajshahi | BD6 | -1.80 | 0.44 | -1.35 |
| Ethiopia (ET) n = 8 | Oromia | ET1 | 0.22 | 0.74 | -2.20 |
| | Oromia | ET2 | -0.48 | 1.66 | -1.93 |
| | Amhara | ET3 | -0.84 | 1.51 | -1.22 |
| | Amhara | ET4 | -0.32 | -0.25 | -1.51 |
| | Oromia | ET5 | -0.24 | 1.04 | -0.14 |
| | Oromia | ET6 | 0.51 | 1.61 | -1.29 |
| | SNNPR[a] | ET7 | 0.01 | 1.39 | -2.68 |
| | SNNPR[a] | ET8 | -0.81 | -0.31 | -3.10 |
| India (IN) n = 12 | Haryana | IN1 | -0.46 | 0.26 | 1.77 |
| | Bihar | IN2 | 0.38 | 1.66 | 1.51 |
| | Uttar Pradesh | IN3 | 0.77 | -2.50 | 1.08 |
| | Madhya Pradesh | IN4 | -1.61 | -2.53 | -0.03 |
| | Bihar | IN5 | -1.77 | -0.98 | 0.60 |
| | Madhya Pradesh | IN6 | 0.58 | -2.30 | -0.06 |
| | Madhya Pradesh | IN7 | 0.25 | -1.68 | -0.27 |
| | Bihar | IN8 | -0.94 | -1.93 | 0.40 |
| | Punjab | IN9 | -0.94 | 0.88 | 2.79 |
| | Uttar Pradesh | IN10 | 0.15 | -1.17 | 1.33 |
| | Bihar | IN11 | -2.25 | -3.55 | 0.02 |
| | Uttar Pradesh | IN12 | -0.36 | -1.49 | 0.86 |
| Malawi (MW) n = 2 | Central Region | MW1 | 0.55 | -1.45 | -1.20 |
| | Central Region | MW2 | 1.00 | -0.74 | -3.18 |
| Mexico (MX) n = 6 | Oaxaca | MX1 | 3.11 | -0.01 | 0.24 |
| | Chiapas | MX2 | 0.79 | 0.14 | 0.39 |
| | Oaxaca | MX3 | 1.03 | -0.78 | -3.26 |
| | Chiapas | MX4 | 2.67 | 0.66 | 1.72 |
| | Chiapas | MX5 | 2.99 | 0.90 | 0.61 |
| | Oaxaca | MX6 | 0.88 | -2.07 | -0.15 |
| Morocco (MO) n = 3 | Fes-Meknes | MO1 | -1.73 | -0.96 | 1.14 |
| | Fes-Meknes | MO2 | -1.13 | -1.95 | 0.96 |
| | Fes-Meknes | MO3 | -1.13 | -2.32 | -0.07 |
| Nepal (NP) n = 6 | Bagmati Pradesh | NP1 | 4.48 | -0.11 | 0.35 |
| | Province No. 5 | NP2 | 2.20 | 0.77 | 1.44 |
| | Gandaki Pradesh | NP3 | -0.10 | 0.36 | 0.26 |
| | Karnali | NP4 | 1.39 | -0.04 | 0.54 |
| | Province No. 5 | NP5 | 0.87 | 0.95 | 1.26 |
| | Gandaki Pradesh | NP6 | 3.18 | 1.58 | 1.62 |

(*Continued*)

**Table 3.** (Continued)

| Country | Region* | Village Code** | Gender Climate | Opportunity | Connectivity |
|---|---|---|---|---|---|
| Nigeria (NI) n = 4 | Plateau State | NI1 | 1.81 | -1.05 | -1.48 |
| | Oyo State | NI2 | 2.14 | 3.62 | 0.32 |
| | Kaduna State | NI3 | 0.88 | 1.76 | 1.36 |
| | Oyo State | NI4 | 2.31 | 1.55 | -1.56 |
| Pakistan (PK) n = 7 | KPK[b] | PK1 | -1.26 | -1.77 | 1.23 |
| | KPK[b] | PK2 | -2.89 | -0.07 | -0.62 |
| | KPK[b] | PK3 | -1.70 | 0.10 | 0.76 |
| | KPK[b] | PK4 | -2.61 | 2.23 | 1.29 |
| | Balochistan | PK5 | -2.16 | -0.56 | 2.33 |
| | Balochistan | PK6 | -1.35 | -0.20 | 0.68 |
| | Sindh | PK7 | -2.16 | -0.42 | -0.54 |
| Tanzania (TZ) n = 4 | Morogoro Region | TZ1 | 1.37 | -1.87 | 1.04 |
| | Arusha region | TZ2 | 1.83 | -0.95 | -0.42 |
| | Morogoro Region | TZ3 | 1.37 | -3.29 | 0.15 |
| | Tanga Region | TZ4 | 1.31 | 0.52 | -0.75 |
| Uzbekistan (UZ) n = 4 | Bukhara | UZ1 | -1.41 | 1.11 | 1.96 |
| | Samarqand | UZ2 | 2.36 | -0.47 | 2.60 |
| | Andijon | UZ3 | 1.58 | 2.26 | 2.85 |
| | Kashkadaryo | UZ4 | 0.86 | 0.71 | 1.32 |
| Zimbabwe (ZW) n = 4 | Masvingo | ZW1 | 3.45 | 0.81 | -1.64 |
| | Midlands | ZW2 | 1.37 | 0.59 | -0.97 |
| | Mashonaland Central | ZW3 | 1.56 | -0.07 | -3.20 |
| | Mashonaland Central | ZW4 | 1.92 | 0.32 | -2.78 |

*Or equivalent to Province or State.

**Codes are used to facilitate analysis as well as to protect the anonymity of the villages surveyed.

[a] Southern Nations, Nationalities, and Peoples' Region.

[b] Khyber Pakhtunkhwa.

relatively more relaxed GC (Fig 3). Most country cohorts appeared rather homogeneous in terms of GC, either in positive (relaxed) or negative (restrictive) ways. For instance, the Bangladesh cohort reported restrictive GC across the 6 villages surveyed (average GC = -1.95, sd = 0.23) whereas the Tanzania cohort exhibited a relatively relaxed GC across the 4 villages part of the study (average 1.47, sd = 0.24).

However, large variability was found in the Uzbekistan and Nepal country cohorts (Fig 3 and Table 3). For instance, one village from the Uzbekistan cohort scored 2.36 (UZ2) while another registered a negative GC score (UZ1 = -1.41). This large variability can be partly explained by estimates of women selling in the local market (75% in UZ2 versus almost none in UZ1); of active female discussants (60% in UZ2 versus 5% in UZ1); as well as women's relatively higher freedom of mobility and increased ability to make strategic life choices in UZ2 compared to UZ1. Similarly, and whereas one Nepali village scored the highest across the dataset (NP1 = 4.48) another scored close to '0' (NP3 = -0.10). Again, this can be partly explained by differences in terms of estimates of women sellers in the local market (almost all in NP1 versus almost none in NP3); control over women's agricultural income (controlled by women *only* in NP1 versus *joint* control in NP3); active female discussants in public meetings and trainings (80% in NP1 versus 16% in NP3); as well as by women's relatively higher freedom of

mobility and increased power and freedom to make life decisions in NP1 *vis-à-vis* NP3. In part (d), we examine these and other issues that further explicate the large variability between the two Nepali villages.

## (b) Complementary components: Opportunity and Connectivity

In addition to the gender data two PCs provided complementary information on the surveyed villages. The first was 'Opportunity'. This component indicated high values for larger villages with better commercial and job training opportunities and low values for smaller villages with less opportunities (Fig 3). Villages with high scores were primarily characterized by their access to (agricultural/non-agricultural) job trainings or vocational programs, high share of households selling their agricultural produce in local markets, high population, and access to a secondary school. Villages with low scores were mainly characterized by lack of access to local markets, health clinics, secondary schools, and to job trainings or vocational programs. Overall, 'Opportunity' registered high heterogeneity within country cohorts probably due to maximum diversity sampling, but also to important economic and demographic differences within the same country areas. Some of these differences are most prominent in terms of access to agricultural and non-agricultural trainings; number of households selling their produce in local markets; and population size. Despite the large variability within country cohorts, on average, the villages with better commercial or job training opportunities were found in Nigeria, Uzbekistan, Afghanistan and Ethiopia while those with lower opportunities were from Morocco, Tanzania, and India (see Table 3).

The second complementary PC was 'Connectivity' which indicated high values for villages with better access to communication and infrastructure services and low values for villages with less access (Fig 3 and Table 3). Villages with high scores were mainly characterized by internet access, the presence of a secondary school, electricity, and a bus line within a 30-minute walk. Villages with low scores reported limited or no access to electricity, upper secondary school or bus line, greater distance to the nearest town with government offices and rapid population growth. On average the study villages from Uzbekistan (average = 2.18, sd = 0.69) and Nepal (average = 0.91, sd = 0.60) appeared to be better connected than villages from Zimbabwe (average = -2.15, sd = 1.03), Malawi (average = -2.19, sd = 1.40) or Ethiopia (average = -1.76, sd = 0.93). Overall, high scores for 'Connectivity' were more associated to good infrastructure than to favourable economic opportunities.

## (c) Relation between components

A pairwise correlation analysis between the two complementary components and the gender climate indicated no statistical relation (r<0.2 and p.value>0.05). This is further confirmed by the visualization of the villages' scores of '*Gender Climate' x 'Opportunity'* and '*Gender Climate' x 'Connectivity'* grouped by country cohort (Fig 4). For instance, on the far left of the spectrum, the Afghanistan cohort exhibited the most restrictive gender climate of the study (average GC = -2.7) despite having three out of the four villages surveyed located in the high quadrant of 'Opportunity'. A subsequent variance partitioning analysis further confirmed the absence of relationship between the three PCs. However, the variance analysis did evidence the strong influence of the local context, which explained up to 76% of the GC variability. Due to this significant result, the next section examines the influence of the local context for agricultural innovation in two study villages.

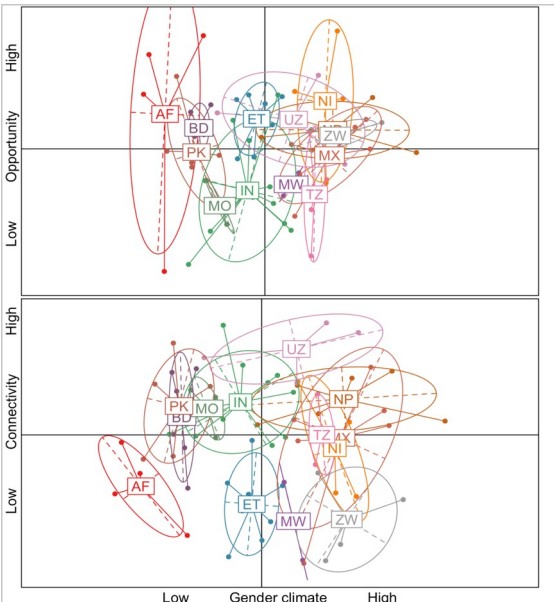

**Fig 4. Visualization of relation between components grouped by country cohort.** The scores of the villages are grouped by country cohort (two letter code). 'Gender climate' is in x-axis, while the two complementary components, 'Opportunity' and 'Connectivity', are represented in y-axis.

## (d) Insights from two villages in Nepal

The Nepal cohort (n = 6) registered the most relaxed GC of our database (average = 2.00). However, it also reported large variability (sd = 1.65). This was especially evident in villages NP1 (GC = 4.48) and NP3 (GC = -0.10) which, interestingly, had relatively similar scores in 'Connectivity' (NP1 = 0.35, NP3 = 0.26). A closer examination of these two villages based on their GC variability was conducted to contextualize the results of the correlation and variance partitioning analyses; specifically, the influence of the local context in women's engagement with agricultural innovation.

*Village #1 NP1, Chitwan District*

NP1 is a village in Chitwan district, Bagmati Pradesh. It is located in the Terai-Duar savanna and grasslands region characterized by tall grasslands and tropical savanna climate around 100–300 m above sea level [71]. With an approximate population of 1,000 inhabitants, the village is culturally diverse: with people from Tharu, Brahmin, and Chhetri origins. Agriculture is the main livelihood. Maize farming constitutes one of the most important commercial activities in the village, however, remittances, mostly from men working abroad, are also an important source of income (NP1 FGDs).

Traditionally grown for home-consumption, maize as a cash crop is relatively new in NP1. Improved maize varieties were first brought into the village in 2011 by two landlords who were part of a local farmers' group mostly comprised of men (NP1 Community Profile; FGDs). Nonetheless, commercial maize farming today is a women-led activity (NP1 FGDs). Furthermore, as pointed out by a man farmer, the first person in the village who planted improved varieties in a field was a woman (NP1 FGDs). Spear-headed by women, key informants indicated that maize production has increased: initially 3–4 bags of maize were produced from 7 *kattha* land (aprox. 0.2 ha) whereas now 22–23 bags can be obtained from the same land (NP1 Community Profile). Currently most agribusinesses in the village are led by women; however, this was not the case two decades ago, as noted by a woman farmer: "earlier women were

regarded as dust particles of men's legs" (NP1 FGD). Women reflected on their capacity to make life and work decisions back then: "earlier men used to manage all the financial expenses and make decisions. There were many restrictions for women. We were not allowed to go out-. . .we were confined within the house. . .to do household activities. Not any women groups and associations were there during that time" (NP1 FGDs). The importance of women's associations to trigger positive gender normative changes in the village was also recognized in the focus group discussions with men:

> During 1996–97 a women's group was formed. In that time a gender training was held. . . [and after that] all the mothers organized a movement to make the village free of alcohol. Though, the men had tried to block that movement they couldn't stop it. Now, the village is free from alcohol and the rate of domestic violence has also reduced. From the past ten years to date we don't have that much violence. If anyone perpetrates violence against women, the women's group and the youth club caught him and punish him. *(NP1 FGDs)*

Women further recognized the role of women's associations in the shift from maize as a subsistence or home-consumption crop to a commercial crop; particularly via the formation of women's cooperatives which allowed women to access credit, agricultural machines (such as tractors and rotavators) and other inputs (e.g., hybrid maize and fertilizer) and further encouraged woman-to-woman peer support in agricultural matters (NP1 FGDs). Besides women's groups, study participants noted other factors including: (1) the expansion of the poultry industry in Chitwan which created a market for maize; (2) high male outmigration which opened opportunities for women to become directly involved in agricultural processes; (3) government and NGOs education programs to promote gender equity and women's empowerment; (4) relative increase and access to agricultural trainings offered by the government and NGOs working in the area; and (5) the construction of new roads which facilitated trading (NP1 FGDs; Community Profile). Despite these advances, women farmers spoke about difficulties to further develop their agribusinesses. Specifically, the scarcity of information in their village about new technologies and practices, and issues related to seed access, quality and availability (NP1 FGDs).

*Village #2 NP3, Myagdi District*

NP3 belongs to Myagdi district, Gandaki Pradesh. Located in the Middle Hills in a semi-forested area (300–3000 m above sea level), the village is part of the Community Forestry Program of Nepal which regulates access and use of natural resources in the region [72]; NP3 Community Profile). Approximately 1,200 people live in the village, the majority being from Chhetri descent. Remittances constitute the main source of income; however, maize farming is also important in this village as it is grown for household consumption and animal feed (NP3 Community Profile).

Similar to NP1, the village also reported high male outmigration as well as increased levels in education, improvements in infrastructure and the presence of women's groups (NP3 Community Profile). Women in NP3 noted that, while women's groups are not necessarily focused on agricultural issues, they "are playing a great role these days to bring all village women together to understand each other better" (NP3 FGDs). Likewise, the new road and transport service has helped to connect their village with the municipal city (NP3 Community Profile). Although these developments were welcomed, focus group discussions with women indicated that some of the potential benefits for agricultural innovation were hindered by relatively restrictive gender norms in terms of (1) access to agricultural knowledge, (2) access to key inputs for agriculture, (3) women's decision-making power, and (4) women's mobility. For instance, whereas key (male and female) informants described women in NP3 as generally

"more into farming than men" (NP3 Community Profile); women in the focus groups noted their limited knowledge about agricultural technologies as well as their relatively lower capacity to decide about agricultural matters. As pointed out by a woman farmer: "it's mostly men who take the decisions, or we take them together as a family, but we [women] do not know anything about improved. . .seeds and we do not have information regarding new seeds from the agriculture office" (NP3 FGDs). A lack of information on sustainable agronomic practices could partly explain women's perceptions regarding improved seeds: "if we grow hybrid seeds then we have to use chemical fertilizer and use of chemical fertilizer is prohibited in our village because it makes the soil hard" (NP3 FGDs). Women also discussed limitations in terms of access/control to capital and general restrictions to their mobility, as noted by another woman farmer: "female innovators do not possess money and cannot go anywhere like male innovators" (NP3 FGDs). With no market in the village and without being able to travel to city markets, women farmers had resorted to other strategies. For instance, one woman successfully negotiated with a middleman from the city to come to the village and purchase her agricultural products (NP3 FGDs).

Despite the comparatively more restrictive (gender) normative climate in this village *vis-à-vis* NP1, women acknowledged that normative change is happening: "earlier women were confined within four walls of the house. . .because of development, changes came here, and we became more aware. . .but still women cannot come forward to speak" (NP3 FGDs). This was further echoed in focus group discussions with men: "the awareness of women has raised [because] when their husbands are out [working abroad] they have had to deal with all the work in the household and outside, which has made them stronger and slowly made them earn a living" (NP3 FGDs).

## Discussion and reflections

This article set to empirically test the relationship between gender norms and economic, infrastructure, and population conditions which the literature recognizes as key for agricultural innovation. Anchored in the notion of *gender climate* we proposed a novel, interdisciplinary, mixed-method approach to identify patterns and differences across villages within specific geographies (i.e., country cohorts) to evidence local conditions that can inform the planning and targeting of AR4D interventions. We tested this approach using gender-related, demographic, infrastructure, and economic data from 70 villages across 13 countries.

Overall, the analysis empirically demonstrates the lack of relationship between 'Gender Climate' and 'Opportunity'/'Connectivity'. The results suggest that enabling conditions for agricultural innovation as those identified under 'Opportunity' (e.g., agricultural and non-agricultural trainings or farmer organizations) or 'Connectivity' (e.g., closeness to markets or access to public services) do not suffice to explain the complexity of agricultural processes. Notably, the variance analysis evidenced the strong influence of the local context—accounting for 76% of the GC variability. Moreover, our results indicate that even when enabling conditions for agricultural innovation linked to 'Opportunity' or 'Connectivity' are present in a rural village, these conditions do not necessarily translate into relaxed gender climates that facilitate gender-inclusive innovation. A deeper examination of two case-study villages from Nepal contextualized the results, most notably with women farmers from NP3 noting that increased education or recent investments in roads and transport did not necessarily translate in gender normative relaxation in terms of women's decision-making or in freedom of mobility. The lack of statistical relation between the two complementary components and the 'Gender Climate' as well as the insights from the two case-study analyses suggest the need to identify gender normative conditions *together* with economic, demographic, infrastructure and agroecological elements in the

targeting of agricultural interventions to better comprehend how the local context influences innovation processes. Overall, our results indicate that the conditions encompassed under the three components ('Gender Climate', 'Connectivity' and 'Opportunity') are *complementary* in their contribution for creating an enabling environment for gender-inclusive innovation. However, they might not be enough. Importantly, the two case studies suggest that not only do gender norms need to be considered as a research dimension in equal pairing with the economic, infrastructure, population or agroecology dimensions in the planning and targeting of AR4D interventions; but, other elements related to the local context such as government regulations and cultural issues might also be relevant. This latter point resonates with several works that stress the importance of including the political and sociocultural context in interventions on agricultural innovation, see for instance [8, 23, 30, 37, 59].

## Opportunities for AR4D interventions

Our results can contribute to inform and sharpen the strategies of AR4D interventions in relation to project design and/or operationalization—including in the identification of clear goals and objectives according to local considerations. Likewise, the empirical findings can be used as a baseline to assess, monitor, and evaluate gender-inclusive progress within or across villages or regions. For instance, interventions could set more realistic goals based on the information herewith presented and maximize their chances of gender equitable outcomes and benefit-sharing by specifically targeting areas within a region or a country where gender norms are more supportive for both women and men.

Similarly, the integration of gender climate considerations into the targeting of AR4D could facilitate the development of diverse measures to support gender-transformative change in different types of local gender climates. For instance, the gender-related statistical results from the Nepal cohort as well as the two case-study villages highlighted opportunities for AR4D interventions in terms of project design, operationalization, and evaluation. AR4D projects could strategize ways of furthering the already existing agri-entrepreneurial mindset of the women in the two case-study villages as expressed by their engagement with maize farming (NP1) or by their manoeuvring of normative mobility restrictions (NP3). The relative importance of commercial maize in NP1 as well as the shift in gender normative realities in this village offer agricultural interventions at least two opportunities: (i) (re)direct and adapt training strategies in maize-related technologies and practices from a focus on men to also include women farmers; and (ii) invest in value chain development, including strengthening linkages with local seed companies. Considering the relatively more restrictive gender climate in NP3 as well as its agro-ecological context, AR4D interventions could resort to relational approaches such as family methodologies to ensure that entire families discuss and co-design suitable alternatives to develop their village. Training for and education in sustainable practices would be especially relevant bearing in mind NP3's forestry and soil conservation regulations.

Because our approach considers economic, infrastructure and demographic conditions (encompassed under 'Connectivity' and 'Opportunity') the interventions could further assess how important is the presence or absence of these conditions for their targeting strategy. That is, how relevant is for a village or study site to have a secondary school, electricity, a bus line within a 30-minute walk, internet or access to (agricultural/non-agricultural) trainings in the operationalization and achievement of gender-inclusive innovation processes? As evidenced in NP3, an intervention focused on maize farming in the Middle Hills in Myagdi District, Nepal would probably have to look beyond whether the prospective villages are well-connected to markets and cities or whether the general population has access to agricultural information

to also consider the (gender) normative structures that enable women and men to equally access and benefit from these developments.

Finally, our statistical gender findings confirm some of the issues emphasised by literature on women's empowerment. Particularly, they echo qualitative and quantitative evidence generated from decades of research in gender and agriculture including the importance of women's access to key agricultural inputs such as credit, technologies, or agricultural trainings. Our gender findings are novel; however, because the results are not based at the individual or household level but at the village level. This allows AR4D interventions to have a broader understanding on the prevailing gender normative conditions that may affect agricultural innovation in a specific location. Likewise, the results also evidence that across villages—although to varying degrees—gender norms linked to decision-making and mobility are particularly important for women to engage in agricultural innovation processes. This suggests that, regardless of where the AR4D intervention is located, if the aim is to become gender-inclusive the intervention will need to develop strategies to account for women's mobility and decision-making issues.

## Opportunities for future research

Given the innate complexity of gender norms' assessment as well as the limited literature in this regard—particularly in agriculture—there remain many opportunities for future research. We highlight three of them. *First*, future work on mixed methods focused on the *gender climate* would benefit from expanding/refining the variables to also include relevant information that captures within-village heterogeneity, i.e., the complex intersectional doings that drive the fluidity of norms and that could influence the agricultural innovation capacity of women and men from distinct backgrounds (e.g., in terms of age, marital status, caste, religion, etc.). For instance, the study could ask: what are the gender norms that regulate the agricultural innovation opportunities of younger women/men in these villages, and how are they different or similar to those governing the opportunities of older women/men? *Second*, replication of our approach in the same or in different villages could benefit from larger samples within a specific geography to explore more broadly the shared gendered patterns as well as the other (non-gendered) local village specificities. *Third*, our use of mixed (numerical and qualitative) data derived from focus group discussions and community profiles allowed for statistical analyses as well as for the contextualization of two case-study villages. However, future studies could use our statistical results to conduct in-depth studies in one or more villages to confirm, reject or nuance the findings. For instance, and whereas our statistical findings overall suggested a relatively homogenous gender climate in Tanzania and Nigeria, qualitative studies conducted in these same villages found important gender and social differences across sites that could impact future gender-inclusive agricultural interventions in these areas—see [15] for Tanzania and [73] for Nigeria.

## Conclusion

Interventions on agricultural innovation are embedded in diverse and complex social contexts that influence the distributional effects as well as the kinds of impacts achievable. However, research informing the planning and targeting of agricultural interventions remains primarily focused on agroecological conditions, market access and/or population information with few studies also looking at social or gender normative conditions. We have advanced a replicable approach that integrates key local conditions (including on gender norms, demographics, infrastructure, and economics) to inform the design and targeting of gender-inclusive innovation in 70 rural villages across 13 countries. Overall, our statistical and case-study findings

suggest that investments in economic, infrastructure, or other rural developments do not necessarily translate into relaxed gender climates that can facilitate gender-inclusive innovation. This indicates that gender normative issues need to be fully integrated and explicitly considered in the targeting and planning strategies of agricultural research for development interventions. Based on our results, we have also highlighted some opportunities for AR4D interventions to integrate gender normative issues in project design, operationalization and evaluation to facilitate the development of diverse measures to support gender-transformative change in different local gender climates.

As gender transformative methodologies in agri-food systems develop, research to identify and capture normative changes at household, village, region, and country levels will become more relevant. This article and its accompanying tutorial constitute an initial step towards the development of what we envisage will soon constitute a key aspect of gender work in agriculture.

## Acknowledgments

The authors wish to thank the women and men from the 70 rural villages who shared their valuable time, knowledge, and experiences so that this research could happen. We are also very grateful to Patti Petesch and Margreet van der Burg for their valuable input in previous versions of this article and to Dina Najjar for allowing us to use her data from Morocco and Uzbekistan. We also appreciate Gideon Kruseman's insightful comments about targeting of AR4D. This research was carried out within the auspices of Wageningen University & Research, The Netherlands.

## Author Contributions

**Conceptualization:** Diana E. Lopez, Lone B. Badstue.

**Data curation:** Diana E. Lopez, Romain Frelat.

**Formal analysis:** Diana E. Lopez, Romain Frelat.

**Methodology:** Diana E. Lopez, Romain Frelat.

**Software:** Romain Frelat.

**Supervision:** Lone B. Badstue.

**Visualization:** Romain Frelat.

**Writing – original draft:** Diana E. Lopez, Lone B. Badstue.

**Writing – review & editing:** Diana E. Lopez, Romain Frelat, Lone B. Badstue.

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
