## [Decision Letter · Decision Letter 0]

30 Dec 2021

PONE-D-21-14940Towards gender-inclusive innovation: assessing local conditions for agricultural targetingPLOS ONE

Dear Dr. Diana E. Lopez,

Thank you for submitting your manuscript to PLOS ONE. After careful consideration, we feel that it has merit but does not fully meet PLOS ONE’s publication criteria as it currently stands. Therefore, we invite you to submit a revised version of the manuscript that addresses the points raised during the review process.

We look forward to receiving your revised manuscript.

Kind regards,

Carlos Alberto Zúniga-González, Ph.D

Academic Editor

PLOS ONE

Journal Requirements:

"The authors acknowledge the financial support of the CGIAR Programs on MAIZE and WHEAT (grant number 2100723600). The views expressed in the article are those of the authors and not of any organization."

"D.E.L

The lead author acknowledges the financial support of the CGIAR Programs on MAIZE and WHEAT (grant number 2100723600). URL for MAIZE: https://maize.org/; URL for WHEAT:https://wheat.org/ 

4.Please amend the manuscript submission data (via Edit Submission) to include author Romain Frelat and Lone. B. Badstue.

Additional Editor Comments:

Dear authors, in order to improve the quality of your manuscript, I suggest you listen to the reviewers' suggestions. And attend the suggested references.

References

1. Zuniga González, C. A., & Villanueva, J. L. J. (2012). Wages and Employs for Non-Farm Agricultural Activities: One Livelihood Strategy in Nicaragua. Global Journal of Management And Business Research, 12(15). Available at: <https: 784="" article="" gjmbr="" index.php="" journalofbusiness.org="" view="">

2. Zúniga-González, C. A., Jarquín-Saez, M. R., Martinez-Andrades, E., & Rivas, J. A. (2016). Participative action Research: A Generation Knowledge approach. Ibero-American Journal of Bioeconomy and Climate Change, 2(1), 218–224. https://doi.org/10.5377/ribcc.v2i1.5696

Reviewers' comments:

Reviewer's Responses to Questions

 **Comments to the Author**

1. Is the manuscript technically sound, and do the data support the conclusions?

Reviewer #1: No

Reviewer #2: Yes

2. Has the statistical analysis been performed appropriately and rigorously? 

Reviewer #1: Yes

Reviewer #2: Yes

3. Have the authors made all data underlying the findings in their manuscript fully available?

Reviewer #1: No

Reviewer #2: Yes

4. Is the manuscript presented in an intelligible fashion and written in standard English?

Reviewer #1: Yes

Reviewer #2: Yes

5. Review Comments to the Author

Reviewer #1: The scientific evaluation of the article with the gender perspective has a global impact, it will serve as the basis for other investigations because it has a relational character with the dimensions and variables which involve cultural, economic, political aspects, among others. Next, it is observed that the research used a mixed methodology, which involved a technological and descriptive processing of experiences originating from semi-structured interviews. Finally, the conclusions propose that this initiative requires more research studies derived from this process, in addition, gender normative need to be integrated into the strategies and objectives of AR4D to ensure livelihoods and food security, particularly for women.

Reviewer #2: When compared to the original manuscript, there is a significant improvement. However, the conclusion focused on the findings rather than how applicable they were to the labor market and the industry. Proofreading is also required to correct a few mistakes in the work.

6. PLOS authors have the option to publish the peer review history of their article (what does this mean?). If published, this will include your full peer review and any attached files.

Reviewer #1: No

Reviewer #2: No

While revising your submission, please upload your figure files to the Preflight Analysis and Conversion Engine (PACE) digital diagnostic tool, https://pacev2.apexcovantage.com/. PACE helps ensure that figures meet PLOS requirements. To use PACE, you must first register as a user. Registration is free. Then, login and navigate to the UPLOAD tab, where you will find detailed instructions on how to use the tool. If you encounter any issues or have any questions when using PACE, please email PLOS at figures@plos.org. Please note that Supporting Information files do not need this step.</https:>

---

## [Author Response · Author response to Decision Letter 0]

25 Jan 2022

Please see document attached 'Responses to Reviewers'. We have addressed all points raised by the Editor and Reviewers and provide our detailed point-by-point responses in the document.

---

## [Editor Report · Decision Letter 1]

27 Jan 2022

Towards gender-inclusive innovation: assessing local conditions for agricultural targeting

PONE-D-21-14940R1

Dear Dr. Diana E. Lopez,

We’re pleased to inform you that your manuscript has been judged scientifically suitable for publication and will be formally accepted for publication once it meets all outstanding technical requirements.

Kind regards,

Carlos Alberto Zúniga-González, Ph.D

Academic Editor

PLOS ONE

Additional Editor Comments (optional):

Dear authors, I have reviewed that you have corrected the comments of the reviewers and I would like to congratulate you for the effort you have made to improve the quality of your manuscript.
---

## [Editor Report · Acceptance letter]

14 Mar 2022

PONE-D-21-14940R1 

Towards gender-inclusive innovation: assessing local conditions for agricultural targeting 

Dear Dr. Lopez:

I'm pleased to inform you that your manuscript has been deemed suitable for publication in PLOS ONE. Congratulations! Your manuscript is now with our production department. 

Kind regards, 

on behalf of

Dr. Prof. Carlos Alberto Zúniga-González 

Academic Editor

PLOS ONE